# Mechanisms of sterilizing immunity provided by an HIV-1 neutralizing antibody against mucosal infection

Elie Richel[1], Arne Cordsmeier[1], Larissa Bauer[1], Kirsten Fraedrich[1], Ramona Vestweber[2], Berit Roshani[2], Nicole Stolte-Leeb[2], Armin Ensser[1], Christiane Stahl-Hennig[2◦], Klaus Überla ®[1◦]*

1 University Hospital Erlangen, Institute of Clinical and Molecular Virology, Friedrich-Alexander Universität Erlangen-Nürnberg, Germany, 2 German Primate Center, Göttingen, Germany

◦ These authors contributed equally to this work.
* klaus.ueberla@fau.de

**Data Availability Statement:** The raw data supporting the findings described in this paper are available in the Supplementary Raw Data file. Further information and inquiries for resources and

## Abstract

Broadly neutralizing antibodies (bnAbs) against HIV-1 have been shown to protect from systemic infection. When employing a novel challenge virus that uses HIV-1 Env for entry into target cells during the first replication cycle, but then switches to SIV Env usage, we demonstrated that bnAbs also prevented mucosal infection of the first cells. However, it remained unclear whether antibody Fc-effector functions contribute to this sterilizing immunity. Therefore, additional challenge viruses were produced that contain SIV Env and graded doses of a fusion-defective trimer of HIV-1 Env, to which the bnAb, PGT121 can bind without interfering with the SIV Env-based cell entry. After administration of either PGT121 or its mutant deficient in Fc-effector functions, rhesus macaques were intrarectally exposed to these challenge viruses and to those using either HIV-1 Env or SIV Env for entry into the first cells. Both antibodies similarly reduced infection events with the challenge virus using HIV-1 Env by a factor close to 200. Incorporating fusion-defective HIV-1 Env trimers into the particles of the challenge viruses at densities observed in primary virus isolates did not reduce SIV Env-mediated infection events. The results indicate that the sparsity of bnAb binding-sites on HIV-1 virions limits the contribution of Fc-effector functions to provide sterilizing immunity against mucosal viral infection. Hence, harnessing Fc-effector functions for sterilizing immunity against mucosal HIV-1 infection may require strategies to increase the degree of antibody opsonization.

## Author summary

Control of the HIV pandemic remains a world-wide health challenge. Although antibodies against HIV have been shown to protect from HIV infection, the precise step during mucosal acquisition of HIV infection and the mechanisms involved are not fully understood. Using a unique infection model in non-human primates that assesses the efficacy of antibodies in preventing infection of the first cells we demonstrate that a non-

reagents should be addressed to the lead contact, Klaus Überla (klaus.ueberla@fau.de). Reasonable requests for reagents for scientific purposes from non-profit organizations will be fulfilled using standard MTAs.

**Funding:** This work was supported by grants from the Deutsche Forschungsgemeinschaft to C.S-H. (STA 447/6-2) and K.Ü (UE 45/13-2). The funders had no role in study design, data collection and analysis, decision to publish, or preparation of the manuscript.

**Competing interests:** The authors declare that no competing interest exists.

neutralizing antibody that binds to the challenge virus without blocking viral entry into the cell does not protect. Under neutralizing conditions, the same antibody prevented infection of the first cells independent of its interaction with Fc-receptors required for non-neutralizing mechanisms of protection. Therefore, we conclude that harnessing non-neutralizing mechanisms of anti-HIV antibodies to provide sterilizing immunity may require strategies to improve the density of antibodies binding to the surface of HIV particles.

## Introduction

Neutralizing antibodies frequently correlate with protection from virus acquisition and disease progression [1]. Therefore, the adoptive transfer of neutralizing monoclonal antibodies is a valuable strategy for the treatment and prevention of viral infections, particularly in immuno-compromised individuals [2–4]. In the case of human immunodeficiency virus type 1 (HIV-1), immune responses induced by infection or vaccine candidates explored so far do not provide protection from infection or disease progression due to a number of viral immune escape mechanisms and the diversity of the circulating HIV-1 variants [5]. However, the identification and isolation of potent broadly neutralizing antibodies (bnAbs) targeting the HIV Envelope (HIV Env) have been a breakthrough for novel prophylactic and therapeutic strategies against HIV/AIDS [6,7]. The efficacy of HIV bnAbs has been thoroughly investigated and protection from systemic infections with chimeric Simian-Human Immunodeficiency Viruses (SHIV) was observed in non-human primates [8–10]. In addition, a prophylactic clinical trial with the bnAb VRC01 provided evidence for protection from acquisition of VRC01-sensitive HIV isolates, but did not reduce the overall HIV-1 infection rate [11]. Therefore, other bnAb alternatives and combinations are currently being investigated. Among them, the V3 glycan-dependent PGT121 bnAb, described to protect against virus acquisition in pre-clinical studies [12], displayed safe and well tolerated pharmacokinetic properties in a phase 1 clinical trial [13].

Although there is strong evidence that administration of bnAb can confer protection against viral acquisition through neutralization, the difficulties in inducing broadly neutralizing antibody responses by vaccination including the lack of binding of stabilized HIV Env trimers to the germ-line precursor antibodies of the bnAbs [14–16] has led to investigations on the relevance of non-neutralizing effector mechanisms. An early study using the first-generation b12 bnAb indicated that Fc-effector functions contributed to protection against virus challenge in non-human primates since an Fc-effector function-deficient antibody, the b12[LALA] mutant, displayed reduced protection compared to the parental antibody [17]. However, more recent reports highlighted that administration of the PGT121[LALA] variant was as efficient as PGT121 at protecting macaques against repeated challenges [18,19]. In addition, entirely abrogating the Fc-effector functions of PGT121 through the LALA-PG mutation did not reveal substantial differences in efficacies compared to the parental antibody either [19]. Hence, the relevance of Fc-effector functions of neutralizing antibodies for protection from virus acquisition remains to be elucidated to determine if efforts should be directed to the induction of non-neutralizing antibodies.

HIV-1 acquisition, as measured by the absence of viremia and lack of serum conversion, can be blocked prior to infection of the first cells by virions present in the contagious material an individual is exposed to, by elimination of the infected first cells, or by containment of the virus and infected cells at a later step at the portal of entry or the draining lymph node

(discussed in [20]). Blocking infection of the first cells, here defined as sterilizing immunity, may be particularly important for antibody-based prevention strategies against HIV-1 to reduce the risk of reactivation of virus from latently infected first cells after the administered antibodies have declined below protective levels. In a previous study of our group, we explored whether bnAbs against HIV-1 can prevent infection of the first cells after mucosal exposure to a challenge virus that uses HIV-1 Env for entry during the first replication cycle, but then switches to SIV Env usage [20]. Since the HIV-1 bnAbs used do not bind to SIV Env, the strong reduction of infection events observed in this study can only be due to blockage of infection events prior to entry into the first cells. Additionally, non-neutralizing effector mechanisms seemed to contribute to the sterilizing immunity provided by the bnAb PGT121, since PGT121 prophylaxis also reduced the number of infection events with an SIV challenge virus that incorporated membrane-anchored HIV gp120 in addition to a functional SIV Env mediating the first entry event. Because PGT121 binds to the surface of this challenge virus without interfering with the SIV Env-mediated entry process, non-neutralizing effector mechanisms should be responsible for the reduction of the number of infection events observed. As already pointed out in the previous study, one limitation was that the number of gp120 molecules of the challenge virus exceeded the number of trimers on the surface of primary isolates of HIV-1 by far. Thus, it was not possible to exclude that an artificially high degree of opsonization of the gp120-containing challenge virus by PGT121 favored the effectiveness of non-neutralizing Fc-dependent effector mechanisms. Therefore, we now performed a comprehensive study by i) using switching challenge viruses containing SIV Env and graded doses of a fusion-deficient HIV Env trimer and ii) investigating the degree of sterilizing immunity provided by a PGT121 mutant lacking Fc-effector functions. Neither approach provided evidence for a role of non-neutralizing effector mechanisms in preventing infection of the first cells.

## Results

### Development of switching challenge viruses differing in the number of PGT121 binding sites

We previously developed SIV-based, Env switching challenge viruses in which the SIV Env gene of the proviral DNA is inactivated by a short out-of-frame duplication (SIVdup) [20]. Cotransfection of SIVdup with expression plasmids for HIV Env results in pseudotyped challenge viruses that use HIV Env for entry during the first infection cycle. In the course of reverse transcription, the short out-of-frame duplication is removed in approximately 30% of the first infection events activating expression of SIV Env and resulting in the outgrowth of SIV. To further investigate the role of Fc-effector functions for sterilizing immunity and the effect of the degree of virion opsonization, we sought to generate switching challenge viruses [20] pseudotyped with SIV Env and presenting varying amounts of a fusion-deficient HIV Env (fdEnv) at densities similar to the ones observed in patient isolates. To mimic natural conditions as closely as possible, we also chose to incorporate a trimeric, membrane-anchored, conformationally stabilized Env, namely ConSOSL.UFO.750, which has been designed to favor bnAb binding [21], into of the particles of our switching challenge virus. As controls, switching challenge viruses were also generated that were only pseudotyped by fully functional HIV Env or SIV Env, respectively.

In the absence of any treatment, all pseudotyped challenge viruses should use their respective Env to infect the first cells and then convert to SIV Env usage for all following rounds of replication (**Fig 1A**). In contrast, prophylactic administration of the bnAb PGT121 or its LALA-PG mutant, both binding to HIV Env but not to SIV Env, should inhibit viral entry of the HIV Env but not SIV Env pseudotyped challenge viruses (**Fig 1B**). Analyzing whether SIV

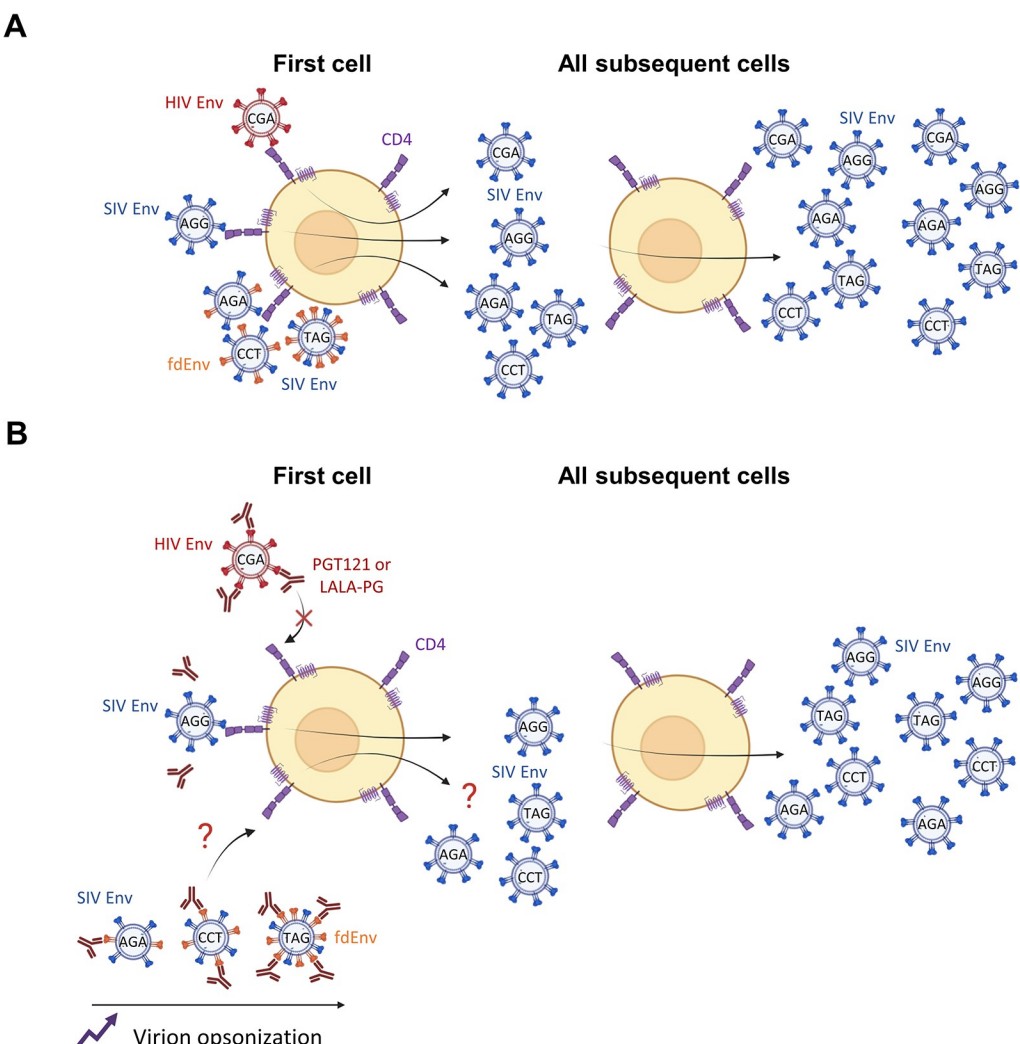

**Fig 1. Design of NHP experiments with Env switching challenge viruses. (A)** Conversion of switching challenge viruses in the absence of bnAbs. All SIVdup pseudotyped particles use their respective Envs to infect the first cell but then switch to SIV Env usage for the following rounds of replication. Each challenge virus possesses a unique genetic tag that is maintained in the viral genome after switching to SIV Env usage. **(B)** Potential outcomes of challenge with the switching viruses in the presence of PGT121 and its LALA-PG mutant. As PGT121 only binds to HIV Env and not SIV Env, inhibition of entry should occur before the first round of infection. The inhibition of the fdEnv challenge virus would provide evidence of non-neutralizing mechanisms contributing to sterilizing immunity and different fdEnv densities could reveal whether the number of antibody binding sites affect the degree of sterilizing immunity. Created with BioRender.com.

Env pseudotyped challenge viruses presenting different amounts of fdEnv during the first round of infection are also inhibited by PGT121 should further reveal the role of non-neutralizing mechanisms in sterilizing immunity (**Fig 1B**).

To produce switching challenge viruses incorporating both SIV Env and varying degrees of fdEnv, HEK293T/17 cells were transiently co-transfected with SIVdup, SIV Env and increasing amounts of fdEnv expression plasmids. Following purification by ultracentrifugation through 35% sucrose cushions, the respective pseudotyped particles, SfdEnv$^{Low}$, SfdEnv$^{Inter}$ and SfdEnv$^{High}$, were analyzed by Western blot. SIV Env, fdEnv and SIVp27 were readily detectable (**Fig 2A**). Importantly, transfection of HEK293T/17 cells with different amounts of the

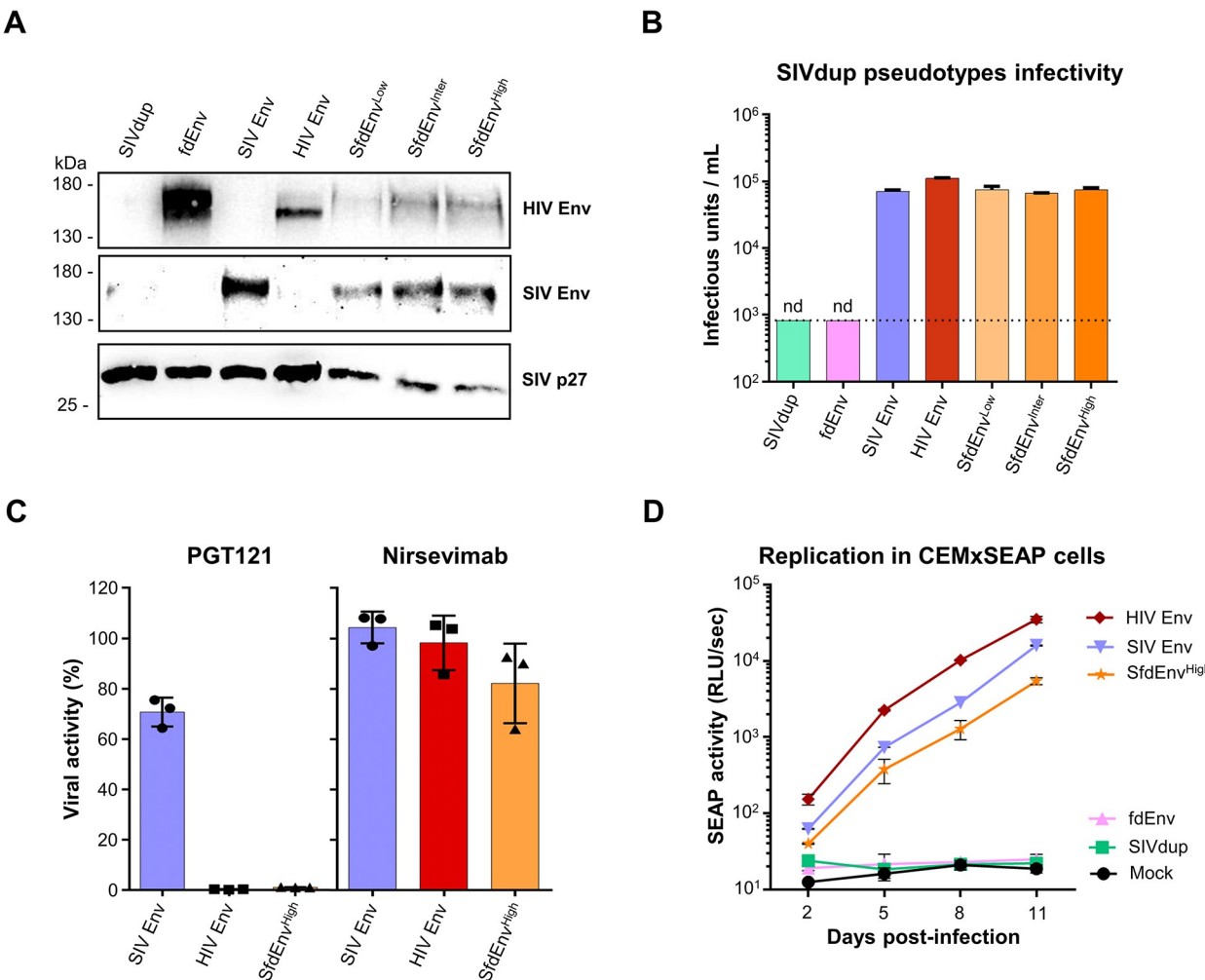

**Fig 2. Characterization of switching challenge viruses.** **(A)** Western blot analyses of particles generated by co-transfection of SIVdup with the indicated Env expression plasmids. SfdEnv[Low], SfdEnv[Inter] and SfdEnv[High] correspond to SIVdup pseudotyped particles that co-incorporated SIV Env and a low, intermediate or high quantity of fusion-defective Env, respectively. **(B)** Infectious titers of SIVdup particles on TZMbl cells. One out of two independent experiments is displayed. Mean and SD from duplicates are shown. The dotted line corresponds to the detection limit of the assay and values below the detection level are indicated with nd. **(C)** Depletion of infectious pseudotyped particles by PGT121-coated beads. Dynabeads protein G previously coated with PGT121 or anti-RSV-F (Nirsevimab) antibodies were incubated in triplicates with the indicated pseudotyped particles and then depleted by magnetic separation. The loss of infectious particles compared to non-treated samples was measured by luciferase activity on TZMbl cells. Bars represent mean and SD of the experimental triplicates. The overlaid data points display the mean of triplicate measurements on TZMbl cells for each experimental triplicate. **(D)** Replication kinetics of the indicated SIVdup pseudotypes in CEMxSEAP reporter cells. The infectious dose was adjusted to $2\times10^4$ IU. Mock condition consisted of cells cultured in R10 medium only, while non-pseudotyped SIVdup particles were used as undiluted supernatant of transfected HEK293T/17 cells. Mean and SD of SEAP activity from the infected cells was measured in duplicates for each time point. One representative experiment out of two is shown.

fdEnv expression plasmid resulted in varying degrees of fdEnv in the pelleted particles, suggesting that fdEnv incorporation onto SIVdup pseudoviruses could be modulated. As expected, SIVdup particles pseudotyped with the fdEnv alone were non-infectious on TZMbl cells, as SIVdup particles lacking any form of Env (**Fig 2B**). Infectious viruses were obtained after pseudotyping SIVdup particles with both fdEnv and SIV Env at levels comparable to HIV and SIV Env pseudotypes lacking fdEnv.

Since purification of virions by ultracentrifugation on sucrose cushions does not discriminate viruses from exosomes, incorporation of SIV Env and fdEnv into SIVdup particles was

confirmed by co-immunoprecipitation of virions using PGT121-coated protein G beads. As expected, SIVp27, SIV Env and fdEnv were detected in the elution fraction from the immuno-precipitated SfdEnv$^{High}$ particles (**S1 Fig**). PGT121 immunoprecipitation also resulted in coprecipitation of SIV p27 in case of HIV Env, but not SIV Env pseudotypes. Thus, these results indicate that fdEnv and SIV Env are both incorporated in SIVdup particles after co-transfection of the respective expression plasmids into HEK293T/17 cells. To exclude that SfdEnv$^{High}$ particles contain a substantial proportion of pseudotypes that contain SIV Env, but no fdEnv, the infectious titers of SIVdup pseudotypes were determined with and without deple-tion by PGT121-coated beads. Depletion with PGT121 beads reduced the infectious titers of HIV Env and SfdEnv$^{High}$ pseudotypes by a factor close to 455 and 98, respectively (**Fig 2C**). In contrast, depletion with beads coated with an irrelevant control antibody or depletion of SIV Env pseudotypes with PGT121 beads only had minor effects on the infectious titers (**Fig 2C**).

To validate that the SIVdup particles pseudotyped with both fdEnv and SIV Env lead to rep-lication-competent SIV, CEMxM7-R5 cells were infected with the SIVdup pseudotypes and virus replication was monitored by co-culture with CEMxSEAP reporter cells. The cultures were passaged every 2–3 days post-infection and secreted embryonic alkaline phosphatase (SEAP) activity in the cell supernatants of the infected cells was measured for each time point. As anticipated, SIVdup pseudotyped particles with HIV Env, SIV Env and SfdEnv$^{High}$ dis-played rapidly increasing SEAP activities. Due to differences in the entry efficacy mediated by HIV Env and SIV Env in CEMxM7-R5 cells, different SEAP activities were observed at day 2. Thereafter, all viruses seemed to replicate with comparable kinetics, as expected from switch-ing of the challenge viruses to SIV Env (**Fig 2D**). Importantly, SEAP activity from the cells incubated with SIVdup alone or fdEnv pseudotyped particles remained comparable to non-infected cells, consistent with the lack of infectivity of those particles as demonstrated for TZMbl cells.

Altogether, these results indicate that the co-transfection of SIVdup, SIV Env and fdEnv expression plasmids into HEK293T/17 cells leads to the production of SIVdup particles incor-porating SIV Env and fdEnv on their surface which can be modulated by varying the amount of fdEnv expression plasmids transfected. Moreover, pseudotyping SIVdup particles with fdEnv alone did not lead to replication-competent viral particles while co-incorporation of fdEnv and SIV Env into SIVdup particles conferred infectivity through SIV Env activity.

To prepare for challenge experiments in rhesus monkeys, the SfdEnv$^{Low}$, SfdEnv$^{Inter}$ and SfdEnv$^{High}$ challenge viruses were characterized for the degree of incorporation of fdEnv into SIVdup particles. Initially, the pseudoviruses were purified by iodixanol velocity gradients to minimize the amount of exosomes from the preparations [22,23]. The iodixanol gradient frac-tions displaying the highest infectious events on TZMbl cells were pooled and analyzed by quantitative ELISAs for SIVp27 and fdEnv (**S2 Fig**). After conversion into molarities, the num-ber of fdEnv molecules per particle were calculated with the assumption that 2,000 molecules of SIVp27 are incorporated per SIVdup particle [24,25]. The quantities of fdEnv trimer mole-cules per virion of SfdEnv$^{Low}$, SfdEnv$^{Inter}$ and SfdEnv$^{High}$ pseudotyped particles were 3.1, 6.9 and 23.5, respectively (**Table 1**), closely mirroring results obtained for primary HIV-1 isolates [26,27]. Consistent with similar infectious titers (**Fig 2B**), no major differences were observed for incorporation of SIV Env by the additional incorporation of SfdEnv (**Fig 2A**).

## Binding and neutralizing activities of PGT121 and its LALA-PG mutant

To explore whether Fc-effector functions are relevant for providing sterilizing immunity against mucosal HIV infections, we additionally generated an Fc-effector function-deficient PGT121 variant for further comparison to the parental PGT121. Therefore, the well-described

**Table 1. Determination of the mean of fdEnv trimers on SIVdup challenge viruses.**

|  | SfdEnv$^{Low}$ | SfdEnv$^{Inter}$ | SfdEnv$^{High}$ |
|---|---|---|---|
| Monomeric fdEnv mass concentration (μg/mL) | 0.5 | 1.8 | 3.7 |
| SIV p27 mass concentration (μg/mL) | 15.9 | 29.2 | 17.9 |
| Trimeric fdEnv molarity (nM) | 0.9 | 3.7 | 7.8 |
| SIV p27 molarity (nM) | 588.5 | 1,080 | 660.9 |
| Number of trimeric fdEnv molecules per particle | 3.1 | 6.9 | 23.5 |

L234A/L235A/P329G (LALA-PG) mutations impairing both FcγR binding and complement activation for the human IgG1 isotype [28], were introduced into the sequence of PGT121 antibody. Binding of PGT121 and PGT121$^{LALA-PG}$ to Env transfected HEK293T/17 cells was analyzed by flow cytometry. As expected, both PGT121 variants showed specific binding to HIV Env and fdEnv expressing cells but not to SIV Env-transfected cells (**Fig 3A**). Furthermore, PGT121 efficiently neutralized HIV Env pseudotypes of SIVdup, while SIV Env and SfdEnv$^{High}$ pseudotypes were not neutralized even at a concentration of 100 μg/mL (**Fig 3B**). To determine whether insertion of the LALA-PG mutation affects PGT121 binding activity towards HIV Env, fdEnv-transfected cells were stained with serial dilutions of each PGT121 variant and polyclonal anti-human IgG antibodies coupled to FITC. The flow cytometry analysis revealed that PGT121$^{LALA-PG}$ was as efficient as the parental PGT121 at binding fdEnv expressing-cells in a dose-dependent manner (**Fig 3C**). Additionally, the neutralization activity of both antibody variants was assessed by neutralization assay against SIVdup particles pseudotyped with HIV Env on TZMbl cells. Consistent with the binding assay, PGT121 and PGT121$^{LALA-PG}$ antibodies displayed similar neutralization activity against HIV Env pseudotypes, with IC50s in the range of 19.33±0.33 ng/mL and 19.34±0.35 ng/mL, respectively (**Fig 3D**).

Thus, these results indicated that PGT121 has neutralizing activity only against SIVdup pseudotyped particles with HIV Env and did not bind to SIV Env, while introducing the LALA-PG mutation into PGT121 did not affect the binding nor the neutralizing activities against HIV Env.

## In vitro Fc-effector functions of PGT121 and its LALA-PG mutant

To further investigate the impact of the LALA-PG mutation on PGT121 Fc-effector functions, HEK293T/17 cells-transiently transfected with expression plasmids for rhesus macaque FcγRI, FcγRIIa, FcγRIIIa or cynomolgus macaque FcγRIIb were stained with serial dilutions of each PGT121 variant. Antibody binding to the respective FcγR-expressing cells was detected via AlexaFluor647-conjugated goat polyclonal anti-human Lambda light chain antibodies to avoid binding competition for the FcγRs. The flow cytometric analysis revealed that the insertion of the LALA-PG mutation into PGT121 abrogated binding to all macaque FcγRs in comparison to the parental antibody (**Fig 4A**). Furthermore, a rhesus C1q deposition assay confirmed that PGT121$^{LALA-PG}$ variant also had abolished activity for the classical complement activation pathway (**Fig 4B**).

## Evaluation of sterilizing immunity by PGT121 under neutralizing and non-neutralizing conditions

To control for interindividual variabilities in the susceptibility of outbred non-human primates to mucosal infection, we previously established and validated a simultaneous challenge model with three different switching challenge viruses, that were differentially tagged by genetic identifiers [20]. Since all switching challenge viruses revert to the same SIV (with the

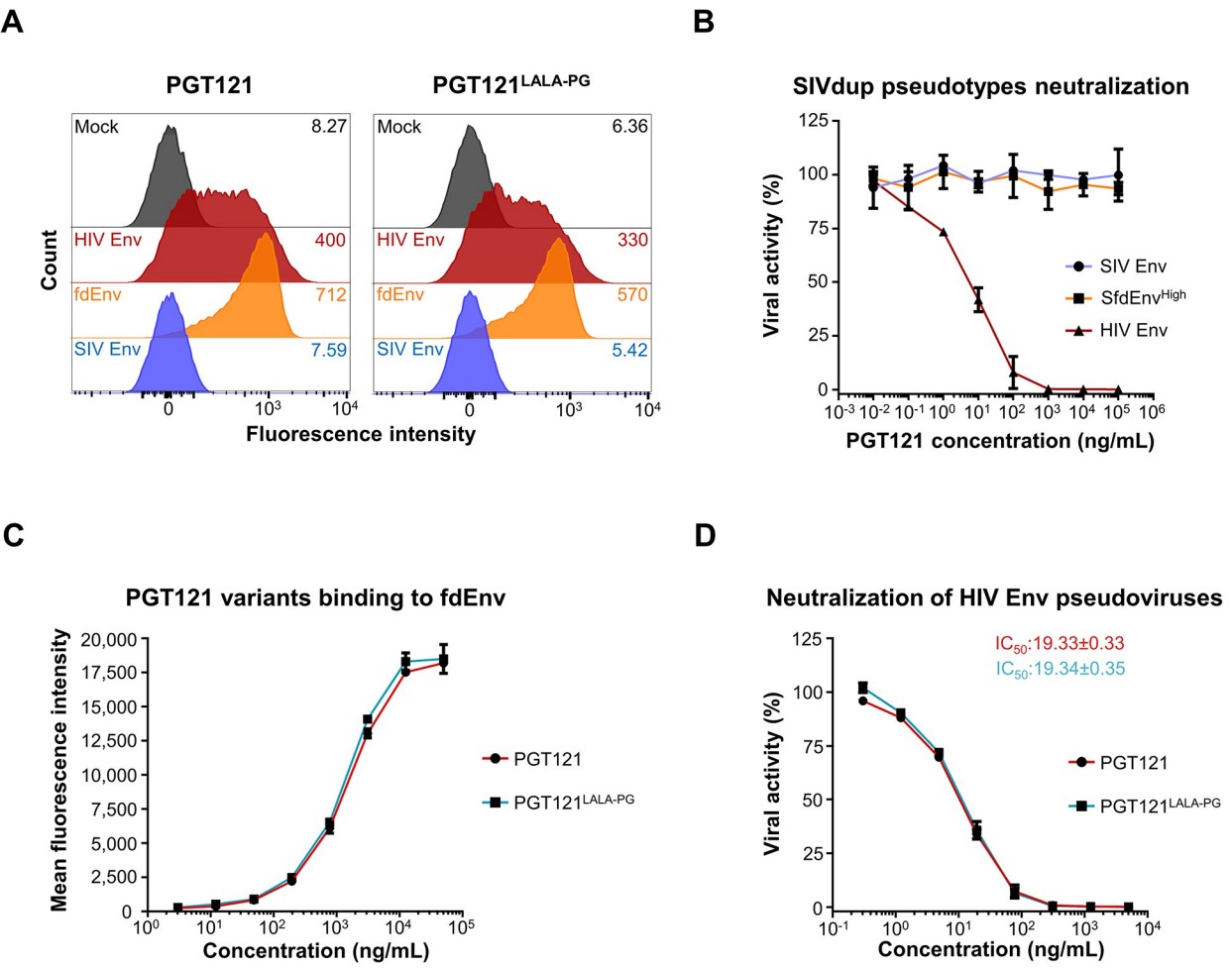

**Fig 3. *In vitro* binding and neutralization activities of PGT121 and PGT121<sup>LALA-PG</sup>.** (A) Binding of PGT121 and PGT121[LALA-PG] to HEK293T/17 cells transfected with either an empty vector (= Mock), HIV Env, fdEnv or SIV Env expression plasmids. The mean fluorescence intensity of the cell population is displayed on the top right corner of each panel. One representative experiment out of two is shown. (B) Neutralization of SIVdup particles pseudotyped with either SIV Env, HIV Env or SfdEnv[High] by PGT121 on TZMbl cells. Mean and SD of two independent experiments each done in duplicates are shown. (C) Dose-response curve of binding of PGT121 and PGT121[LALA-PG] to fdEnv-expressing cells. Mean of duplicates and SD of one representative experiment out of two is shown. (D) Dose-response curve of neutralization of HIV Env pseudotyped SIVdup particles by PGT121 and PGT121[LALA-PG]. The IC50s of the antibodies are displayed on the top right corner of the graph. Mean and SD of two independent experiments each performed in triplicates are shown.

exception of the genetic tag), the ratio of these viruses should reflect the ratio of the first infection events leading to a productive infection. Using this approach a simultaneous challenge experiment was designed with an SIV Env pseudotype, an HIV Env pseudotype and three SfdEnv pseudotypes differing in their density of fdEnv (**Table 1**). For that purpose, each pseudovirus was generated with a unique genetic tag differing from all other tags by at least two nucleotides. The tags consisted of mutations in the wobble position of SIV Gag codons, enabling the detection of the switched challenge viruses by Next-Generation Sequencing (NGS) of PCR amplicons. Since the SIV Env pseudotyped challenge virus should not be affected by PGT121 antibodies, we determined the ratio of the reads of the HIV Env and fdEnv containing viruses relative to the reads of the SIV Env pseudotyped virus. Comparing these ratios between the mock-treated control group and the groups treated with PGT121 antibodies allows to determine the fold-reduction of infection events with HIV Env and fdEnv containing viruses by the respective antibodies.

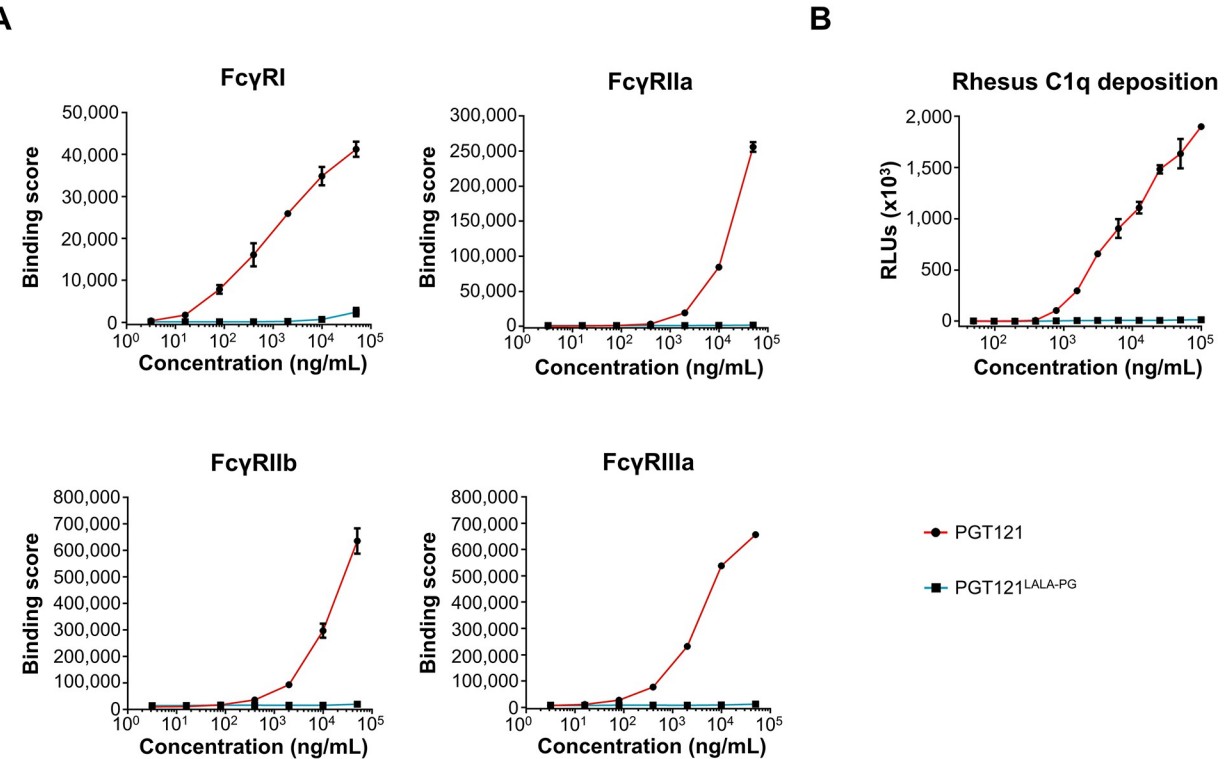

**Fig 4. Fc-γ receptor binding and C1q recruitment by PGT121 and PGT121[LALA-PG].** (A) PGT121 and PGT121[LALA-PG] binding to HEK293T/17 cells transiently transfected with the indicated FcγRs of macaques. Mean of duplicates and SD of one representative experiment out of two are shown. (B) Deposition of C1q of rhesus macaques on PGT121 and PGT121[LALA-PG]. ELISA plates were coated with serial dilutions of the two antibodies and subsequently incubated with 10% rhesus macaque serum. After washing, C1q deposition was detected using a C1q antiserum. Mean of duplicates and SD of one representative experiment out of two is displayed.

The challenge experiment was then conducted by passively administering groups of rhesus monkeys (n = 6 per group) *i.v.* with either a saline solution (Mock), 5 mg/kg body weight of PGT121 (PGT121), 5 mg/kg body weight of PGT121[LALA-PG] (PGT121[LALA-PG]) or a low-dose of 1 mg/kg PGT121 (PGT121-LD) (**Fig 5A**). Seven days later, all animals were challenged intrarectally with a mixture of SIVdup particles pseudotyped with HIV Env ($2x10^5$ IU), SIV Env ($5x10^4$ IU), SfdEnv[Low] ($9x10^4$ IU), SfdEnv[Inter] ($5x10^4$ IU) and SfdEnv[High] ($2.5x10^5$ IU). On the day of the simultaneous challenge, median PGT121 concentrations in the sera of animals from the PGT121, PGT121[LALA-PG] and PGT121-LD groups were 68.6 μg/mL, 62.7 μg/mL and 9.6 μg/mL, respectively (**Fig 5B, S3 Fig**). No statistically significant difference was observed between the PGT121 and PGT121[LALA-PG] treated groups. After the simultaneous challenge all animals should develop viremia since the SIV Env pseudotyped challenge virus is not targeted by PGT121. As anticipated, no major difference in plasma viral load was observed between each experimental group and all animals had detectable viral RNA copies on days 7 and 10/11 post-challenge (**Fig 5C**).

After amplification of plasma viral RNA by RT-PCR and genomic viral DNA from three different lymphatic tissues (mesenteric lymph node (LN), inguinal LN and submandibular LN) by PCR, the number of genetic tag reads derived from each challenge virus was determined at day 10/11 post-exposure by NGS analysis of SIV gag amplicons spanning the tagged sequence (**S1–S4 Tables**). The ratio of the read counts of the genetic tags of HIV Env and fdEnv containing challenge viruses relative to the genetic tag of the SIV Env challenge virus was calculated for at least three different samples from each animal to explore potential

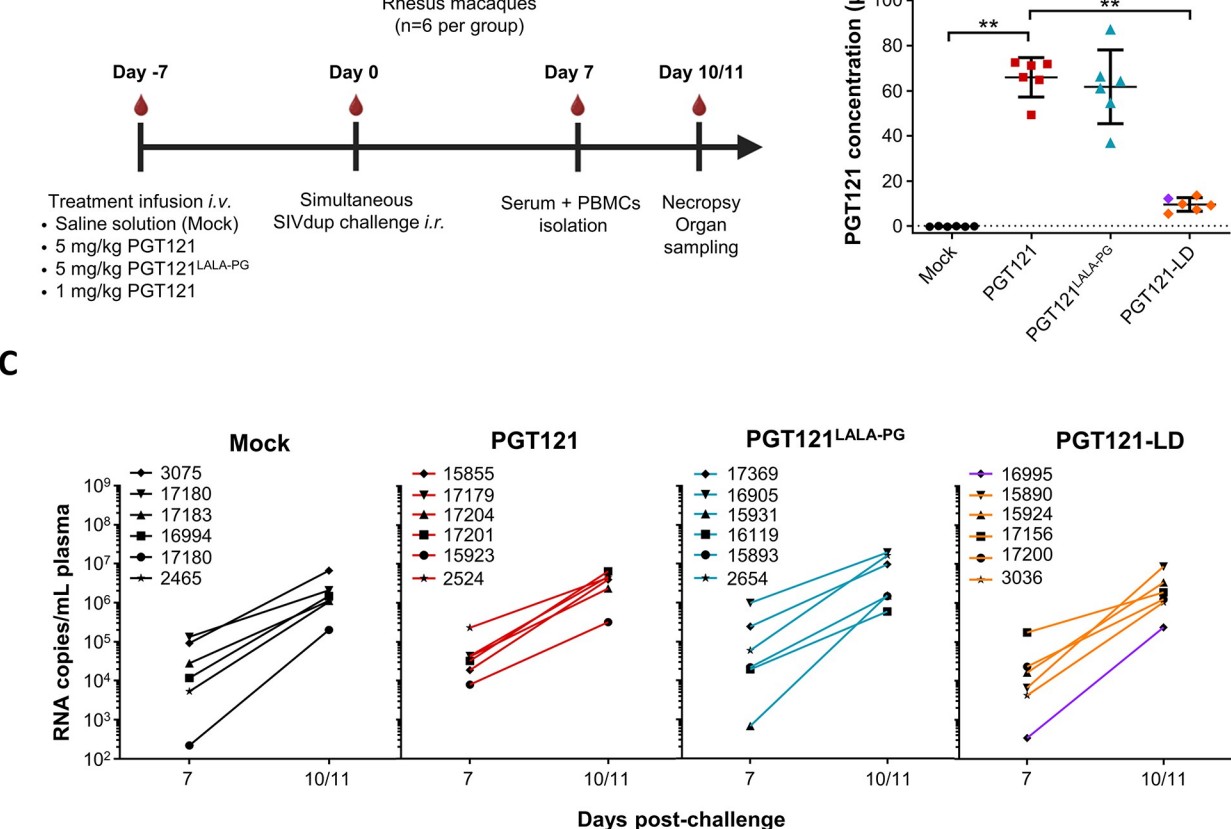

**Fig 5. Simultaneous challenge experiment in non-human primates.** (**A**) Scheme of experimental schedule. Six rhesus macaques per group were infused intravenously (*i.v.*) either with a saline solution (mock), 5 mg/kg PGT121 (PGT121), 5 mg/kg PGT121$^{LALA-PG}$ (PGT121$^{LALA-PG}$) or a low-dose of 1 mg/kg PGT121 (PGT121-LD) seven days prior to intrarectal (*i.r.*) simultaneous challenge with HIV Env, SIV Env, SfdEnv$^{Low}$, SfdEnv$^{Inter}$ and SfdEnv$^{High}$ pseudotypes. Blood samples were collected on day 7 post-challenge and lymphoid organs were sampled after euthanasia of the animals on day 10 to 11 post-challenge. Created with BioRender.com. (**B**) PGT121 serum concentrations on the day of challenge. Bars represent mean and SD of the groups. The individual data points correspond to the mean of PGT121 serum concentrations measured for each animal from two independent experiments each performed in duplicates. The dotted line represents the cut-off value of the ELISA. Statistically significant differences were determined by a two-tailed Mann-Whitney test between each respective group (** $p < 0.005$). (**C**) Viral RNA load in plasma on days 7 and 10/11 post-challenge. Mean of duplicates are shown for each animal. The animal designation numbers are displayed on the top left corner of each graph. Viral loads of the experimental groups did not differ significantly from the viral load of the mock control group (two-tailed Mann-Whitney test, $p > 0.05$). Values for animal #16995 are highlighted in purple.

variations within individual animals (**S4 Fig**). The means of ratios obtained from lymphoid tissues and plasma for each animal of the four experimental groups were combined and then used to calculate medians of the groups, allowing statistical analysis of the influence of PGT121 variants pre-exposure prophylaxis on reducing systemic infection by each challenge virus (**Fig 6**). In comparison to the mock treated group, the ratio of genetic tags derived from HIV Env to SIV Env pseudotypes was reduced by 169 and 212-fold in the PGT121 and PGT121$^{LALA-PG}$ infused animals, respectively, while no substantial difference was detected between these two experimental groups (**Fig 6A**). The median ratio of the HIV Env to SIV Env viruses was also reduced by approximately 60-fold in the PGT121-LD experimental group

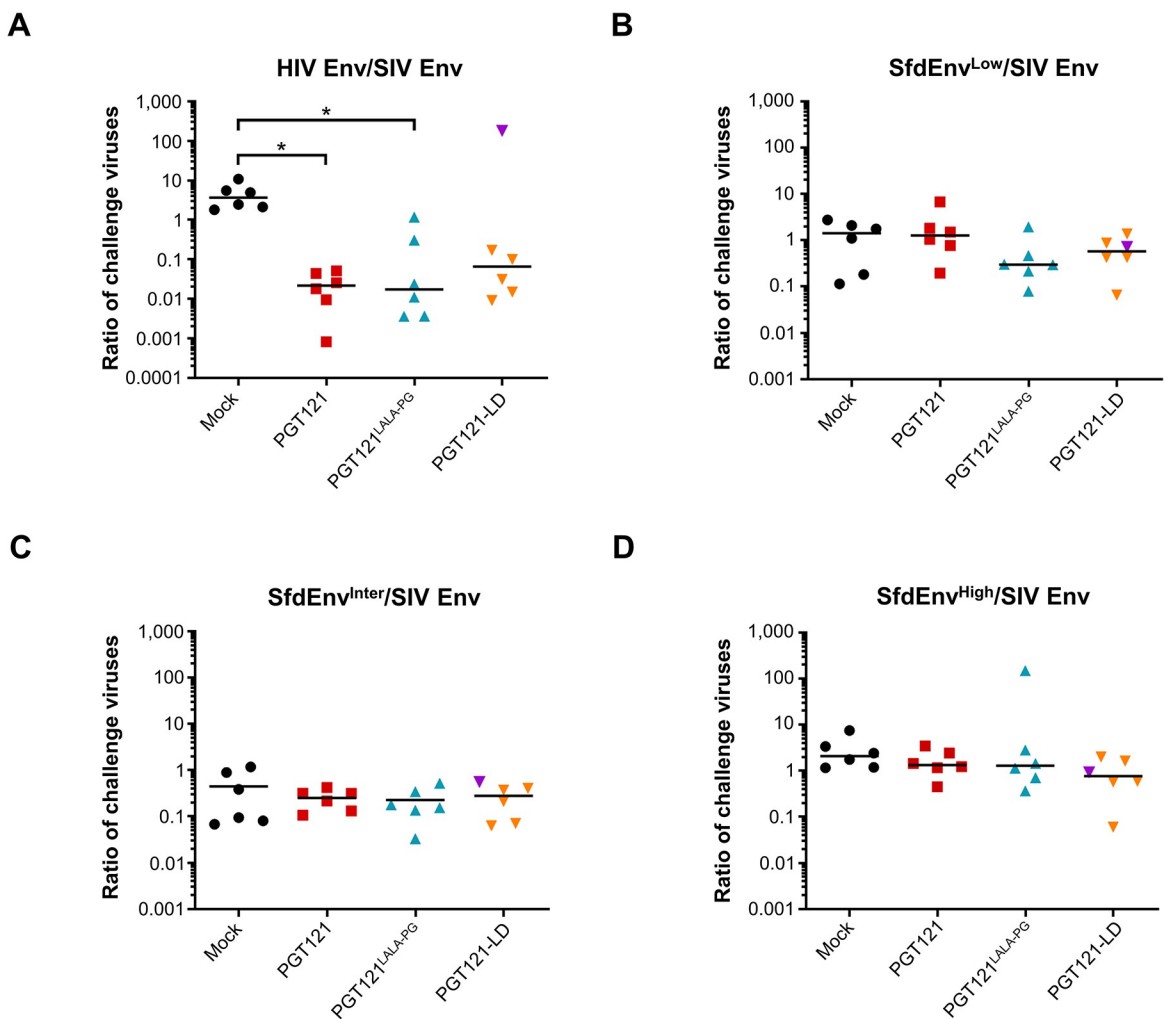

**Fig 6. Ratio of switching challenge viruses.** Mean of ratio of switched challenge viruses derived from pseudotypes with **(A)** HIV Env, **(B)** SfdEnv[Low], **(C)** SfdEnv[Inter], or **(D)** SfdEnv[High] to SIV Env pseudotypes. For each animal (n = 6 per group) the mean ratios of the challenge viruses in plasma and lymph nodes from three different locations (mesenteric, inguinal and submandibular) collected on the day of necropsy are shown as individual data points. Bars represent the median of each group. The ratios for the animal #16995, largely deviating in the HIV Env/SIV Env ratio are marked by a purple inverted triangle symbol. Statistically significant differences between groups as determined by Kruskal-Wallis tests followed by Dunn's multiple comparison tests are indicated (* $p<0.05$).

over the mock-treated animals, but the difference between the groups did not reach statistical significance. One animal from PGT121-LD group (#16995) displayed a high ratio of HIV Env to SIV Env pseudotyped challenge viruses (**Fig 6A**, **S4 Fig**, **S1 Table**), despite a PGT121 serum concentration of 12.2 µg/mL falling within the range of the PGT121-LD-treated group (5.5 to 13.6 µg/mL) (**Fig 5B**, **S3 Fig**). This suggests a particularly high susceptibility of this individual monkey for HIV Env-mediated entry into the first cells. The difference between the PGT121-LD and Mock groups only reached statistical significance after exclusion of the outlier animal (using two-tailed Mann-Whitney test; $p$ = 0.0043).

Importantly, the ratios of the genetic tags of SfdEnv[Low], SfdEnv[Inter] and SfdEnv[High] to SIV Env challenge viruses were not substantially impacted by any of the PGT121 treatments in comparison to the mock group, indicating that simple binding of PGT121 and PGT121[LALA-PG] without neutralization neither suppressed nor enhanced infection of the first cells (**Fig 6B–6D**).

In summary, these results indicate that PGT121 efficiently protects from infection of the first cells by neutralization. Neither impairing PGT121's ability to interact with Fc-γ receptors and recruit the complement system nor exposing physiological levels of binding sites for PGT121 on the surface of the challenge viruses, which can only lead to viral entry inhibition by non-neutralizing mechanisms, affected the number of infection events of the first cells.

## Discussion

Due to the lack of sufficient efficacy of HIV vaccine strategies tested so far, passive administration of bnAbs remains one of the most promising approaches to confer protection against HIV infection (reviewed in [29,30]). Vectored delivery of genes encoding the antibody may also provide long-term protection [31,32]. One concern for such passive immunization strategies against HIV/AIDS is that in the case of non-sterilizing immunity, latently infected cells may be formed from which a systemic HIV infection could be launched once antibody concentrations have declined below protective levels.

PGT121, a bnAb targeting the V3 loop of HIV Env, has been extensively studied and described to provide protection against systemic infection in animal models. Importantly, the study from Stab *et al.* demonstrated that PGT121 confers sterilizing immunity against HIV challenge by inhibiting viral entry into the first cells in non-human primates [20]. Additionally, the study provided initial evidence that non-neutralizing mechanisms contribute to the sterilizing immunity provided by PGT121. The study was based on a switching challenge virus that contained membrane-anchored HIV-1 gp120 and SIV Gag at a molar ratio of 1.8 [20]. As discussed in the previous report, the large number of potential PGT121 binding sites on the virion may have favored efficacy of non-neutralizing effector mechanisms. Since HIV particles of primary isolates have been repeatedly reported to incorporate only 7–14 Env trimer molecules per virion [26,27,33], it remained to be determined whether non-neutralizing effector mechanisms indeed contribute to the sterilizing immunity provided by PGT121.

Hence, our study aimed at generating particles of switching challenge viruses presenting binding sites for PGT121 at more natural densities, which can only lead to viral entry inhibition by non-neutralizing mechanisms. Using challenge viruses presenting 3.1 to 23.5 fdEnv trimers per virion, no evidence for a contribution of Fc-effector function to the sterilizing immunity provided by PGT121 was observed. While concerns could be raised that Fc-effector functions could be masked by a high SIV Env density that negatively interferes with the recruitment of effector cells to PGT121 opsonized virions, the successful depletion of SfdEnv-High particles by PGT121-coated Protein G beads indicates that PGT121 can simultaneously bind to the virion and protein G beads. Therefore, the Fc-fragment of PGT121 bound to virions should also be available for interaction with Fc-γ receptors on the surface of host-immune cells (**Fig 2C**). Furthermore, due to the lateral mobility of lentiviral Env proteins, interfering of SIV Env with the recruitment of effector cells seems unlikely [34]. Importantly, the PGT121[LA-LA-PG] mutant, severely impaired in binding to all macaque FcγRs and in complement deposition, reduced the acquisition of the HIV Env pseudotypes as efficiently as the parental antibody providing independent confirmation that Fc-effector functions do not contribute to prevention of infection of the first cells.

Whether reduction in the number of infection events of the first cells provides sterilizing immunity or not is a matter of challenge dose. To obtain representative results in our simultaneous challenge experiments, high challenge doses were needed. However, we have previously observed prevention of infection after low-dose and repeated-low dose challenge experiments with SIVdup challenge viruses [20]. Since there are only one to five founder viruses after heterosexual transmission [35], reducing the number of first infection events by neutralizing

antibodies by more than 100-fold, as observed in the present non-human primate study, should translate into strong sterilizing immunity in humans as well.

Non-neutralizing effector mechanisms did not contribute to sterilizing immunity provided by PGT121 in our non-human primate challenge model under conditions reflecting more physiological numbers of antibody binding sites on the virion. The challenge virus system used in the present study does not allow to evaluate the role of antibody-dependent cell cyto-toxicity (ADCC), as the infected first cells do not express HIV Env. Defining sterilizing immu-nity as complete prevention of infection of host cells, ADCC-mediated killing of the infected first cells is too late to provide sterilizing immunity in this sense. If sterilizing immunity would be defined as blocking release of infectious viruses from cells of an exposed host, we cannot formally exclude that ADCC could kill the infected first cells prior to release of progeny virus.

The results presented in this report are consistent with the study conducted by Hangartner *et al.* showing that the PGT121$^{LALA-PG}$ mutant confers a similar degree of protection from sys-temic infection with SHIV to the parental PGT121, even at a half-maximal antibody concen-tration [19]. Similarly, passive immunization with PGT121 and PGT121$^{LALA}$ fully protected pigtail macaques from an intrarectal SHIV challenge even in the presence of seminal plasma in the inoculum, which has been shown to inhibit NK cell activation and granulocyte oxidative burst *in vitro* [36]. Additionally, non-human primates were equally protected by PGT121 and PGT121$^{LALA}$ against an intravenous HIV cell-associated virus challenge [18].

Our previous observation that non-neutralizing effector mechanisms provide sterilizing immunity from a challenge virus with a large number of non-neutralizing antibody binding sites on viral particles suggests that increasing opsonization of virions may be a strategy to har-ness non-neutralizing antibodies for sterilizing immunity. Dose-dependent effects of the den-sity of antibody binding sites on *in vitro* Fc-effector functions were observed before. In this case antibody-dependent phagocytosis of HIV-1 particles coated with MPER-specific mono-clonal antibodies was only detectable after increasing MPER antigen coating of virions, further supporting the hypothesis that the natural Env density on HIV-1 particles is insufficient for triggering non-neutralizing effector mechanisms by monoclonal antibodies [37].

Altogether, these findings indicate that non-neutralizing activities of PGT121 neither con-tribute to sterilizing immunity nor to protection from systemic infection. In contrast, administra-tion of the monoclonal antibody b12, a first generation bnAb, was reported to be more effective at protecting against a high-dose vaginal SHIV challenge than a b12$^{LALA}$ mutant [17]. It is conceivable that the intrinsic properties of each antibody, such as the binding site and binding angle, may influence the recruitment of host-effector cells through the engagement of FcγRs. However, *in vitro* comparisons of Fc-effector functions failed to support this hypothesis [19]. Comparing the findings of Stab *et al.*, and our present findings, the number of antibody binding sites on the virus particles seems to affect the susceptibility of the challenge viruses to non-neutralizing effector mechanisms. Therefore, it remains unclear to us, whether differences in Env densities of virions from different preparations of the SHIV challenge viruses used could explain the different outcomes regarding the efficacy of non-neutralizing effector func-tions of PGT121 and b12.

While a maximum of three molecules of a single monoclonal antibody can bind to one HIV Env trimer, polyclonal sera may lead to a higher occupancy and may therefore favor a contri-bution of Fc-effector functions to protection. Several studies investigating the administration of polyclonal non-neutralizing anti-SIV Env antibodies reported modest efficacies against SIV challenge. Although adoptive transfer of non-neutralizing polyclonal anti-SIV Env IgGs from vaccinated monkeys did not confer protection from SIVmac239 acquisition, a significant reduction of viral load in the chronic phase was observed [38]. Furthermore, Alter *et al.* reported that administration of high doses of polyclonal IgGs from immunized macaques

correlated with partial protection from repeated low-dose SIVmac251 challenges in non-human primates despite the lack of detectable neutralizing activities against the challenge virus [39]. Therefore, both studies suggest that virus opsonization or antibody-dependent cytotoxicity by non-neutralizing Env-specific polyclonal antibodies may contribute to protection. Alternatively, bioengineering the Fc-portion to enhance FcγR affinities or complement activation may offer a way to bypass the low number of Env exposed on virions. Although the S239D/A330L/I332E (DEL) mutation had previously been shown to improve binding to rhesus FcγRII and FcγRIII, a recent study conducted by Dias *et al*. did not detect a significant delay in plasma viral load by the DEL-mutated bnAbs compared to the parental antibodies [40]. However, coinciding with the detection of anti-drug antibody responses in NHPs, DEL-mutated bnAbs were also cleared faster and may therefore lack improved efficacy.

In summary, our report demonstrates that PGT121 confers sterilizing immunity against HIV challenge predominantly through neutralizing mechanisms. The low number of Env molecules on the surface of primary HIV isolates seems to be insufficient to trigger protective Fc-dependent effector mechanisms. It remains to be determined, if combining multiple monoclonal antibodies targeting non-competing binding sites results in sufficient opsonization for non-neutralizing effector mechanisms to become efficacious. Furthermore, systematic assessment of the degree of Env densities on the surface of challenge viruses may be necessary to improve the comparability of studies on the importance of Fc-effector functions for protection from HIV infection by anti-HIV Env antibodies.

## Methods

### Ethics statement

The rhesus monkeys (*Macaca mulatta*) used in this report were housed at the German Primate Center (DPZ) and cared for by experienced animal caretakers according to the German Animal Welfare Act complying with the European Union guidelines on the use of non-human primates for biomedical research and the Weatherall report. The animal experimentations were approved by the Lower Saxony State Office for Consumer Protection and Food Safety under the project license 33.9-42502-04-22-00146. In agreement with §11 of the German Animal Welfare act, the DPZ has the permission to breed and house non-human primates under the license 392001/7 granted by the local veterinary office of Göttingen. Throughout the experimentation, monkeys were accommodated in groups of two per cage equipped with a perch, or in single cages with constant visual, olfactory, and acoustic contact with neighboring cages if not socially compatible. In this case, small mash inserts in the cages enabled the separated monkeys to groom with their roommates. Unlimited water was provided, and each animal was fed twice a day with dry monkey biscuits containing appropriate carbohydrate, energy, fat, fiber (10%), mineral, protein, and vitamin content. Fresh fruits or vegetables with treats such as nuts, cereal pulp and different seeds were added to the diet to make foraging more attractive. In addition, the animals received feeding puzzles, toys, and wood stick for entertainment. All along the study, animal caretakers monitored any signs of pain, distress, or sickness by verifying the water and food intake, feces consistency and the general condition of the animals. In case any of the animals presented abnormalities, those were assessed and treated by veterinarians. All animal procedures were performed by qualified veterinarians.

### Generation of uniquely tagged SIVdup expression vectors

The cloning of the genetically tagged SIVdup expression plasmids has been previously described [20]. In addition to the previously generated SIVdup^TAG and SIVdup^CCT plasmids, three additional unique silent mutations differing by at least two nucleotides from each other

were introduced into SIVdup gag sequence by annealing the phosphorylated oligonucleotides pairs dupCCT-AGA-AGG-s (5'GTACAGACAACAGAACCCCATACCCGTCGGTAACAT TTACCGAAGATG) and dupCCT-AGA-AGG-a (5'GATCCATCTTCGGTAAATGTTACCG ACGGGTATGGGGTTCTGTTGTCT); dupCCT-AGG-CGT-s (5'GTACAGACAACAGAA CCCCATACCCGTCGGTAACATTTACAGGCGTTG) and dupCCT-AGG-CGT-a (5'GAT CCAACGCCTGTAAATGTTACCGACGGGTATGGGGTTCTGTTGTCT) or dupAAC-AG A-AGG-s (5'GTACAGACAACAGAACCCCATACCAGTAGGCAACATTTACAGAAGG TG) and dupAAC-AGA-AGG-s (5'GATCCACCTTCTGTAAATGTTGCCTACTGGTATGG GGTTCTGTTGTCT) and ligated into BsrGI/BamHI digested SIVdup vector. The respective SIVdup mutants were designated SIVdup^CCT-CGA, SIVdup^CCT-AGG and SIVdup^AAC-AGA.

## Cell culture

Adherent HEK293T/17 and TZMbl cells were cultured in DMEM supplemented with 10% FCS, 1X Glutamax (all from Gibco, Thermo Fisher Scientific, Waltham, MA, USA) and 1% penicillin/streptomycin (= D10 medium). Suspension CEMxSEAP and CEMxM7-R5 cell lines were cultured in RPMI 1640 (Gibco, Thermo Fisher Scientific, Waltham, MA, USA) also supplemented with 10% FCS, 1X Glutamax and 1% penicillin/streptomycin (= R10 medium). All cell lines were cultured in a humidified incubator at 37°C supplemented with 5% $CO_2$ and passaged every 2–3 days according to cell density.

## Production of SIVdup challenge viruses

SIVdup challenge viruses were produced by transient co-transfection of SIVdup expression plasmids with the respective Env expression vectors into HEK293T/17 cells. One day prior to transfection, $2x10^7$ HEK293T/17 cells were seeded into 175 cm$^2$ flasks in D10 medium. On the day of transfection, 40 μg of uniquely tagged SIVdup expression plasmids were mixed with 20 μg of SIV Env (SIVmac251) or HIV Env (SF162P3Nc8) expression vectors into 5 mL of DMEM + 1X Glutamax (= D0 medium) thoroughly mixed with 180 μg of polyethylenimine (PEI). Following a 20 min incubation at RT, the cell media of the previously seeded HEK293T/17 cells were exchanged for 20 mL of D0 medium, and the transfection reactions were added dropwise onto the cell media. Cells were incubated for 6 h in a $CO_2$ incubator at 37°C and cell media were changed for 30 mL of DMEM + 1.5% FCS + 1X Glutamax and 1% P/S (= D1.5 medium).

To generate SIVdup challenge viruses incorporating SIV Env and varying amounts of fdEnv, 40 μg of uniquely tagged SIVdup expression vectors were mixed with 20 μg of SIV Env expression vector and 20 μg, 10 μg or 2 μg fdEnv plasmids. The DNA content was adjusted to a total of 80 μg per transfection by adding pcDNA3.1 plasmid (empty vector) and mixed with 240 μg PEI. Transfection of HEK293T/17 cells were then carried out as described above.

Subsequently to media exchange, the cells were incubated for 72 h into a humidified incubator supplemented with 5% $CO_2$. The cell media containing the SIVdup challenge viruses were collected and centrifuged at 2,000 x g for 5 min. The supernatants were 0.45 μm filtered and parts of them were aliquoted to be stored at -80°C constituting the virus stocks used for *in vitro* and *in vivo* infection experiments. The remaining part of the supernatants were ultracentrifuged through a 35% sucrose cushion at 28,000 x g for 2 hours 30 min at 4°C using an Optima XPN-80 ultracentrifuge (Beckman Coulter, Brea, CA, USA). The virus pellets were then resuspended into 200 μL PBS and stored at 4°C for further downstream characterization of the SIVdup pseudotyped particles.

## Western blot analysis of pelleted SIVdup pseudotypes

For Western blot analyses, samples of SIVdup pseudotypes pelleted through a 35% sucrose cushion were loaded in reducing conditions onto 10% SDS-PAGE gels, run at 120 V for 1 hour 30 min and transferred onto nitrocellulose membranes. HIV Env and fdEnv were detected via goat polyclonal anti-HIV-1 gp120 antibodies (BP1035, Acris, Herford, Germany), SIV Env with mouse monoclonal anti-SIVmac251 gp120 antibody (KK8, obtained through the NIH AIDS Research and Reagent Program from Dr. K. Kent) and SIV p27 with mouse mono-clonal anti-p24 antibody (183-H12-5C, obtained from the NIH AIDS Research and Reagent Program). Following incubations with the appropriate secondary antibodies coupled to HRP (all from Dianova, Hamburg, Germany), the chemiluminescence induced by the reaction with ECL solution and HRP was captured by the INTAS advanced fluorescence imager (Intas Science Imaging Instruments GmbH, Ahmedabad, Germany).

## Depletion of SIVdup pseudotyped particles by PGT121-beads

To confirm that the majority of SfdEnv pseudotyped particles co-incorporated SIV Env and fdEnv, 30 μL of Dynabeads Protein G (Thermo Fisher Scientific, Waltham, MA, USA) were incubated for 30 min at RT under constant rotation with 50 μg/mL of either PGT121 or anti-RSV-F (Nirsevimab) antibodies diluted in PBS-T. Then, the beads were washed twice with 200 μL PBS following separation using a magnetic rack and resuspended in triplicates per condition with 100 μL of SIV Env, HIV Env or SfdEnv$^{High}$ pseudotypes contained in the cell supernatants of HEK293T/17 cells. After an overnight incubation at 4°C with constant rotation, the beads were separated by a magnetic field and the supernatants transferred to fresh 1.5 mL tubes. Non-treated virus preparations, that were subjected to the same incuba-tion procedures, and the particles-depleted samples were then diluted 1:10 into D1.5 medium, and 50 μL of the dilutions were dispensed in triplicates per condition into a flat-bottom 96-wells plate. Afterwards, 100 μL of a TZMbl cell suspension at 2x10$^5$ cells/mL sup-plemented with 22.5 μg/mL of DEAE-Dextran were dispensed in each well and the cells were incubated for 48 h at 37°C in a humidified incubator with 5% $CO_2$. Finally, the cell media were exchanged for 80 μL of PBS per well and the luciferase activity of the infected cells was measured by an Orion Microplate Luminometer (Berthold Detection System, Pforzheim, Germany) after the addition of 40 μL of Bright-Glo Luciferase assay system (Promega, Fitchburg, MA, USA). The percent of viral activity was then calculated for each experimental triplicates by dividing the luciferase activity of the depleted fractions by the non-treated samples and multiplied by 100.

## Titration of SIVdup pseudotyped particles on TZMbl cells

The determination of the infectious units (IU/mL) of preparations of switching challenge was performed on TZMbl cells carrying a Tat-inducible β-Galactosidase and luciferase expression cassettes as described previously [20]. In short, 1x10$^5$ TZMbl cells were seeded into 24-well plates in D10 medium. A day later, cell supernatants containing the respective SIVdup pseudo-types were diluted 1:10 and 1:100 into D1.5 medium and 250 μL were transferred onto the cells in duplicates. After a 4 h incubation in a $CO_2$ incubator at 37°C, 1 mL of D10 medium was added to each well and the cells were cultured for 48 h at 37°C. Subsequently, the cells were washed with PBS, fixed with 0.5% Glutaraldehyde for 10 min at RT, washed thrice with PBS and incubated for 2 h at 37°C in presence of X-Gal. The titer of infectious particles was finally determined by counting the amount of β-Galactosidase expressing cells in each well using an inverted microscope and expressed in IU/mL.

## Passaging of SIVdup pseudotypes in CEMxSEAP cells

To perform infection kinetics of the SIVdup pseudotypes, $5 \times 10^5$ CEMxM7-R5 suspension cells, expressing both CXCR4 and CCR5 co-receptors, were incubated with $2 \times 10^4$ IU in a final volume of 1 mL of D1.5 medium. Since SIVdup particles' and fdEnv pseudotypes' infectious titers were undetectable on TZMbl cells, 1 mL of the undiluted pseudovirus preparations was used to expose CEMxM7-R5 cells. Similarly, the mock-treated condition consisted of 1 mL D1.5 medium. Following a 4 h incubation at 37°C in a humidified $CO_2$ incubator, the cells were washed thrice with 1 mL of PBS and co-cultured with $1 \times 10^6$ CEMxSEAP cells in 25 cm² flasks in 6 mL of R10 medium. Two days later, 1 mL of the cell suspensions was transferred to fresh $1 \times 10^6$ CEMxSEAP cells in R10 medium and incubated back at 37°C, 5% $CO_2$. The remaining volume of the cell suspensions was centrifuged for 5 min at 2,000 x g, the supernatants were aliquoted in 2 mL tubes and stored at -80°C to later determine the SEAP activity secreted by the infected cells. The same procedure was repeated on days 5, 8 and 11 post-exposure. Finally, the SEAP activity in the cell supernatants was determined for each time point in duplicates using the SEAP Reporter Gene Assay System (Life Technologies, Carlsbad, USA) following the manufacturer's recommendations.

## Determination of fdEnv incorporated per SIVdup particle

To determine the degree of fdEnv incorporation into SIVdup particles, quantitative ELISAs were carried out using an anti-SIV p27 kit (XpressBio, Frederick, MD, USA) and an anti-HIV Env b12 bnAb. Beforehand, the respective pseudotyped particles were separated by iodixanol gradient and the layers displaying the highest infection events on TZMbl cells were pooled together (S2 Fig).

For the fdEnv quantitative ELISA, SIVdup pseudotyped particles with SfdEnv^Low, SfdEnv^Inter and SfdEnv^High were diluted in PBS, distributed into high-binding 96-well plates (Greiner, Kremsünster, Austria), and incubated overnight at 4°C. The next day, the plates were washed thrice with 200 μL PBS-T per well, and 100 μL of mouse IgG2a b12 antibody, diluted at 10 μg/mL in 2.5% powdered milk in PBS-T, were distributed in each well. Following a 1 h incubation at RT, the plates were washed three times with PBS-T and further incubated for 1 h at RT in the dark with goat polyclonal anti-mouse IgG antibodies coupled to HRP (Dianova, Hamburg, Germany) diluted in 2.5% powdered milk in PBS-T. Subsequently, three additional washes with PBS-T were performed and chemiluminescence was measured by a Victor X4 (Perkin Elmer, Hamburg, Germany) after addition of ECL. To assess the amount of fdEnv in virus preparations quantitatively, the ELISA was carried against a standard composed of ConS gp140 CFI [41]. Therefore, taking into account that the molecular weight of fdEnv monomers is 160 kDa, the measured mass concentrations of fdEnv were multiplied by a factor 1.14 (160/140 = 1.14) in order to compensate for the difference in molecular weight between fdEnv and ConS gp140 CFI.

In parallel, SIVp27 concentration was determined using the SIVp27 ELISA kit (XpressBio, Frederick, MD, USA) following the manufacturer's recommendations.

To estimate the number of trimeric fdEnv molecules per SIVdup particles, the measured mass concentrations were converted into molarities considering fdEnv trimers and SIVp27 molecular weights to be 480 kDa and 27 kDa, respectively. Subsequently, the number of fdEnv trimers incorporated per SIVdup particle was calculated with the assumption that each viral particle contains approximately 2,000 SIVp27 molecules:

$$Number\ of\ fdEnv\ trimers\ per\ particle = \frac{Molarity\ of\ fdEnv\ trimers}{\frac{Molarity\ of\ SIVp27}{2,000}}$$

### Recombinant PGT121 antibody variants

Recombinant PGT121 and PGT121[LALA-PG] antibodies were custom-ordered from and produced by Evitria AG (Schlieren, Switzerland). Complementary DNA fragments of the human IgG1 PGT121 sequence [42] were synthetized and cloned into expression vectors. Both PGT121 variants were produced by transfection of suspension-adapted CHO K1 cells, the supernatants were harvested by centrifugation and 0.2 μm sterile-filtered. The antibodies were purified from the supernatants by MabSelect Sure (Cytiva, Marlborough, MA, USA) and solubilized into PBS supplemented with 100 mmol/L L-arginine. Both antibody purity and monomericity were determined to be > 97.5% using size exclusion chromatography with AdvenceBio SEC column 300 A 2.7 μm 7.8x300 mm (Agilent, Santa Clara, CA, USA) and contained less than 1 endotoxin unit per mL as determined by Endosafe PTS system (Charles River, Wilmington, MA, USA). Upon arrival, each antibody was stored at -20˚C in 1 mL aliquots until further use.

### Dose response binding of PGT121 variants to fdEnv by FACS

HEK293T/17 cells co-transfected with Blue Fluorescent Protein (BFP) and fdEnv expression plasmids were incubated in duplicates with 4-fold serial dilutions of PGT121 or PGT121[LALA-PG] starting at 50 μg/mL in FACS buffer. PGT121 binding to fdEnv-expressing cells was detected with goat polyclonal anti-human IgG F(ab')$_2$ fragments coupled to FITC (Jackson ImmunoResearch, West Grove, PA, USA) and cells were fixed with 2% PFA. Samples were acquired on an Attune-NxT Flow Cytometer (Thermo Fisher Scientific, Waltham, MA, USA) and PGT121 binding among BFP-expressing cells was analyzed using FlowJo V10.8.1 software (BD Biosciences, Franklin Lakes, NJ, USA).

### PGT121 neutralization assays

The neutralization potency of PGT121 and PGT121[LALA-PG] were compared by neutralization assay on TZMbl cells as previously described [43]. To this end, 4-fold serial dilutions of the respective PGT121 variant, starting at 5 μg/mL, were made in F-bottom 96-well plates in a final volume of 50 μL in D1.5 medium. Subsequently, 50 μL of SIVdup pseudotyped particles with HIV Env were mixed with the PGT121 dilutions in duplicates. After a 1 h incubation at 37˚C, 100 μL of a TZMbl cell suspension at 2x10$^5$ cells/mL, supplemented with 25 μg/mL DEAE-Dextran, were transferred into each well and incubated for 48 h in a humidified incubator at 37˚C with 5% CO$_2$. The cell media were then exchanged for 80 μL of PBS and after the addition of 40 μL Bright-Glo Luciferase assay system (Promega, Fitchburg, MA, USA) per well, the luciferase activity from the infected cells was measured by an Orion Microplate Luminometer (Berthold Detection System, Pforzheim, Germany). IC$_{50}$ values were calculated using GraphPad Prism 6 (San Diego, CA, USA).

To determine the neutralization activity of PGT121 on SIVdup pseudotypes, the antibody was 10-fold serially diluted into F-bottom 96-well plate in duplicates. SIV Env, HIV Env and SfdEnv[High] pseudotyped particles were then added to the antibody dilutions and incubated for 1 h at 37˚C. Subsequently, 2x10$^4$ TZMbl cell supplemented with 25 μg/mL DEAE-Dextran were transferred into each well and incubated for 48 h in a humidified incubator at 37˚C with 5% CO$_2$. The neutralization assay was then performed as described above.

### PGT121 variants binding to Env-expressing cells

PGT121 and PGT121[LALA-PG] antibodies were diluted at 20 μg/mL in FACS buffer (2% FCS + 2 mM NaN$_3$ in PBS) and incubated with HEK293T/17 cells previously transiently co-transfected

with BFP and either an empty vector (pcDNA3.1), HIV Env, SIV Env or fdEnv expression vectors. Following a 30 min incubation in the dark at RT and 3 washes with FACS buffer, the cells were stained with a mouse monoclonal anti-human IgG antibody coupled to AlexaFluor647 (BioLegend, Hercules, CA, USA) for 20 min at RT in the dark. After three additional washes with FACS buffer, the cells were finally fixed with 2% PFA for 20 min at RT and acquired onto an Attune-NxT Flow Cytometer (Thermo Fisher Scientific, Waltham, MA, USA). Binding of PGT121 variants to the respective transfected cells was analyzed using FlowJo V10.8.1 software (BD Biosciences, Franklin Lakes, NJ, USA) among BFP positive cells.

## PGT121 variants binding to macaques FcγRs

To assess the binding of PGT121 variants to macaque FcγRs, HEK293T/17 cells were transiently transfected with expression plasmids coding for rhesus macaque FcγRI, FcγRIIa, FcγRIIIa and cynomolgus macaque FcγRIIb coupled to GFP-Spark (all from SinoBiological, Beijing, China). Two days later, transfected cells were incubated in duplicates with 50 μL of PGT121 variants 5-fold serially diluted into FACS buffer starting at 50 μg/mL for 30 min at RT in the dark. Subsequent to three washes with FACS buffer, the cells were stained with goat polyclonal anti-human Lambda light chain antibodies coupled to AlexaFluor647 (SouthernBiotech, Birmingham, AL, USA) for 20 min at RT in the dark. Following three additional washes with FACS buffer, the cells were fixed with 2% PFA for 20 min at RT in the dark and acquired onto an Attune-NxT Flow Cytometer (Thermo Fisher Scientific, Waltham, MA, USA). Binding of PGT121 variants to the respective FcγRs among GFP-expressing cells was analyzed using FlowJo V10.8.1 software (BD Biosciences, Franklin Lakes, NJ, USA) and expressed as binding score as described below:

$$Binding\ score = percent\ of\ positive\ cells \times Mean\ Fluorescence\ intensity\ (AF647)$$

## Rhesus macaque C1q deposition on PGT121 variants

High-binding 96-well plates (Greiner, Kremsmünster, Austria) were coated with 2-fold serial dilutions of PGT121 and PGT121$^{\text{LALA-PG}}$ starting at 50 μg/mL in 100 μL PBS per well. The next day, the plates were washed thrice with 200 μL PBS per well and blocked for 1 h at RT with 100 μL 2% BSA in PBS. Following three washes with PBS, 50 μL of rhesus macaque serum (obtained from the German Primate Centre, Göttingen, Germany) diluted 1:10 in DMEM medium were distributed into each well and incubated for 20 min at 37°C. The plates were then washed with 200 μL PBS per well and C1q deposition was detected via sheep polyclonal anti-human C1q antibodies coupled to HRP (BioRad, Hercules, CA, USA) diluted in 2.5% powdered milk in PBS-T and incubated for 1 h at RT in the dark. Finally, the plates were washed thrice with 200 μL PBS-T and chemiluminescence was measured by a Victor X4 multilable plate reader (Perkin Elmer, Hamburg, Germany) after addition of 100 μL ECL in each well.

## Non-human primates experiment

Twenty-five female purpose-bred Indian-ancestry rhesus monkeys (*Macaca mulatta*), of different ages (3.7–15.4 years) and body weights (4.3–9.7 kg), were provided from the breeding colony of the German Primate center. All animals were seronegative for SIV, simian retrovirus type D and simian T-lymphotropic virus type 1. For all experimental procedures, monkeys were anesthetized by *i.m.* injection of a mixture of 5 mg ketamine, 1 mg xylazine and 0.01 mg atropine per kg body weight. Blood samples were collected from the femoral vein using the vacutainer system (BD). Viral challenges were performed intrarectally in a volume of 3 mL as

previously described [44], and animals were kept in ventral recumbency with their hips elevated for 20 min. Animals were euthanized by an overdose injection *i.v.* of 200 mg sodium pentobarbital per kg body weight. Tissue samples were collected at necropsy and consisted of mesenteric, inguinal, axillary, submandibular and rectal lymph nodes as well as spleen, tonsil, and bone marrow.

## Antibody application and simultaneous challenge of SIVdup pseudotypes

Initially one animal was used to decide on the intrarectal challenge doses of each SIVdup pseudotype to be used for the viral mixture of the main study. Based on the proportion of each SIV ´s derived tags in plasma analyzed by NGS, the final doses were calculated to obtain relatively balanced shares of the replicated viruses (see below). Seven days prior to challenge, groups of rhesus monkeys (n = 6 per group) were infused into the saphenous vein with either sterile Dulbecco's phosphate buffered saline w/o Ca++ and Mg++ (Pan-Biotech, Aidenbach, Germany) (= mock-treated), 5 mg/kg body weight PGT121, 5 mg/kg body weight PGT121$^{LALA-PG}$ or a low-dose of 1 mg/kg body weight of PGT121 in a final volume of 10 mL. On day of challenge, each animal was intrarectally exposed to $2x10^5$ IU of HIV Env (CCT-CGA), $5x10^4$ IU of SIV Env (CCT-AGG), $9x10^4$ IU of SfdEnv$^{Low}$ (AAC-AGA), $5x10^4$ IU of SfdEnv$^{Inter}$ (CCT) and $2.5x10^5$ IU of SfdEnv$^{High}$ (TAG) pseudotyped particles. Of note, one animal per experimental group (monkeys #3075, #15855, #17369 and #16995) was inoculated with a mixture of challenge viruses in which the viral stocks had undergone one additional freeze and thaw cycle. Therefore, the infectious doses of each single pseudotype may have been reduced. However, since the ratios of challenge viruses from the mock-treated animal (#3075) did not substantially differ from the other animals among the same group (**S4 Fig, S1–S4 Tables**), the animals were included into the study. Blood samples were collected 7 days post-infection and necropsies were performed 10 to 11 days post-challenge as described above.

## PGT121 serum concentration determination

PGT121 and PGT121$^{LALA-PG}$ serum concentrations were determined by quantitative ELISA in duplicates at day -14 prior to challenge and days 0 and 10/11 post challenge for each animal. After saturation with 5% powdered milk in PBS-T, HIV ConB gp120-coated high-binding 96-well plates (Greiner, Kremsünster, Austria) were incubated with the sera isolated from each monkey diluted at 1:12,500 into 2.5% powdered milk in PBS-T. PGT121 binding was detected via polyclonal anti-human IgG antibody conjugated to HRP (Dianova, Hamburg, Germany) and chemiluminescence was measured by a Victor X4 multilable plate reader (Perkin Elmer, Hamburg, Germany) after addition of ECL. To quantify PGT121 serum concentrations, the ELISAs were carried out against standards composed of either PGT121 or PGT121$^{LALA-PG}$ antibodies.

## Plasma viral load quantification

Viral RNA copy numbers in plasma were quantified essentially as described [20].

## NGS analysis

Following necropsies of the challenged animal on days 10/11 post-infection, viral RNA from plasma (5 animals per group) and viral genomic DNA from mesenteric LN, inguinal LN and submandibular LN (6 animals per group) were extracted using either QIAmp Viral RNA Mini kit or QIAmp DNA Minikit (both from Qiagen, Hilden, Germany), respectively. The gag fragment spanning the SIVdup genetic tags were amplified either by RT-PCR or PCR with the

primers NGS-SIVdup Fw2 (5'TCGTCGGCAGCGTCAGATGTGTATAAGAGACAGCAGG ATCAGATATTGCAGGAACAAC) and NGS-SIVdup Rvs (5'GTCTCGTGGGCTCGGAGA TGTGTATAAGAGACAGTTTACTGCTGCATCTGTCTGTCC). The PCR products were uniquely indexed with dual NEBNext Oligos using the NEBNext ARTIC SARS-CoV-2 Library Prep Kit (both from New England Biolabs, Ipswich, MA, USA) following the supplier's recommendations. After confirming the insertion of the indices at the amplicon's extremities by agarose gel electrophoresis, samples were pooled together, agarose gel purified and resuspended into TE buffer using the NucleoSpin Gel and PCR Clean-up kit (Macherey-Nagel, Düren, Germany) to exclude the excess of undesired products in the preparations.

The amplicon libraries were paired-end sequenced with the MiSeq reagent micro kit v2 (300 cycles) on a MiSeq Instrument (Illumina, San Diego, CA, USA) with read length of 240 and 60, respectively. Sequences were analyzed utilizing CLC GenomicsWorkbench 22 (Qiagen Aarhus A/S, Aarhus, Denmark).

For each sample, the number of sequenced reads corresponding to the SIVdup genetic tags for HIV Env, SfdEnv$^{Low}$, SfdEnv$^{Inter}$ and SfdEnv$^{High}$ pseudotyped particles were divided by the number of reads derived from the SIV Env pseudotypes. The mean of the respective ratios obtained from all biological samples for each animal was calculated and statistically significant differences were calculated between each experimental group by Kruskal-Wallis test followed by Dunn's multiple comparison test using GraphPad Prisms 6.0 (GraphPad Software, Boston, MA, USA).

### Statistical analysis

Statistical differences for PGT121 serum concentrations were determined by two-tailed Mann-Whitney test between each experimental group. Significant reduction of infection of HIV Env, SfdEnv$^{Low}$, SfdEnv$^{Inter}$ and SfdEnv$^{High}$ to SIV Env pseudotypes between the experimental groups was analyzed by non-parametric Kruskal-Wallis test followed by Dunn's multiple comparison test. All tests were performed using the GraphPad Prism 6.0 software for windows. Mean or median, SD and $p$-values are indicated in the respective Figs and Fig legends.

### Supporting information

**S1 Fig. Incorporation of SIV Env and fdEnv into SfdEnv pseudotyped particles.** SIV Env, HIV Env and SfdEnv$^{High}$ pseudotyped particles pelleted through 35% sucrose cushion were subjected to immunoprecipitation by PGT121-coated beads. Dynabeads Protein G were incubated with PGT121 at 50 μg/mL for 30 min at RT under rotation. Subsequent to washing steps, the beads were further incubated with the indicated Virus Preparations (VP) for 30 min at RT under rotation. The Flow Through (FT) fractions were collected by separating the beads using a magnetic rack and transferred to fresh 1.5 mL tubes. Subsequently, the beads were washed thrice with PBS and the elution was performed by incubating the beads at 95˚C for 2 min in presence of 1 X SDS-PAGE loading dye supplemented with 5% β-mercaptoethanol. The VP, FT and Immunoprecipitated (IP) fractions were then analyzed by Western blot for the detection of HIV Env, SIV Env and SIV p27, as indicated on the right side of the Fig. (TIF)

**S2 Fig. Purification and characterization of SfdEnv pseudotyped particles by Iodixanol gradient.** Subsequent to ultracentrifugation on 35% sucrose cushions, the virus fractions (VF) were loaded onto 8% - 24% iodixanol gradient and centrifuged at 250,000 x g for 90 min at 4˚C to separate exosomes and viral particles from the preparations using an Optima XPN-80

ultracentrifuge (Beckman Coulter, Brea, CA, USA). **(A)** Each gradient fraction was analyzed by Western blot for the detection of fdEnv and SIV p27. **(B)** Fractions were diluted 1:10 into DMEM medium supplemented with 1.5% FCS and 1X Glutamax and titrated on TZMbl cells. Two days post-infection, β-galactosidase-expressing cells were counted using an inverted microscope. Bars represent mean of duplicates and SD. Fractions 16% to 22%, displaying the highest infectious events were pooled to further quantify the degree of fdEnv incorporation per particle via quantitative ELISAs.
(TIF)

**S3 Fig. Kinetics of PGT121 serum concentrations.** Rhesus monkeys were either mock-treated or infused *i.v.* with the indicated antibodies 7 days prior to simultaneous challenge with the SIVdup pseudotyped particles. PGT121 serum concentrations were determined for each animal on days -14, 0 and 10/11 post-challenge *via* quantitative ELISA. PGT121 serum concentrations were determined using standards composed of either PGT121 or PGT121$^{LALA-PG}$. The animal designations are displayed on the top left corner of each graph. Data points and bars represent the means and SD of two independent experiments measured in duplicates for each animal.
(TIF)

**S4 Fig. Reads ratio for each animal.** Ratio of switched SIVs derived from SIVdup pseudotyped with **(A)** HIV Env, **(B)** SfdEnv$^{Low}$, **(C)** SfdEnv$^{Inter}$, or **(D)** SfdEnv$^{High}$ to SIV Env in the plasma (red symbols) or from three different lymphoid organs (black symbols). Samples for which no reads of the HIV Env or SfdEnv challenge viruses were detected are marked by blue symbols, the indicated values correspond to the ratio calculated assuming one read and therefore represent the lower limit of detection. Bars mark the medians of the ratios of samples from each animal overlaid with the individual data points of the ratios for each sample.
(TIF)

**S1 Table. Number of reads derived from the different challenge viruses for the mock-treated animals.**
(DOCX)

**S2 Table. Number of reads derived from the different challenge viruses for the PGT121-treated animals.**
(DOCX)

**S3 Table. Number of reads derived from the different challenge viruses for the PGT121$^{LALA-PG}$-treated animals.**
(DOCX)

**S4 Table. Number of reads derived from the different challenge viruses for the PGT121-LD-treated animals.**
(DOCX)

**S1 Text. Methods underlying the supporting data.**
(DOCX)

**S1 Data. Data underlying the findings for this paper.**
(XLSX)

## Acknowledgments

We would like to thank Philipp Brytzki, Sandra Heine, and Nicole Leuchte (German Primate Center, Göttingen, Germany) for excellent technical assistance. We also thank Dr. Robin

Shattock (Imperial College, London, UK) for the kind provision of the ConSOSL.UFO.750 expression plasmid. ConS gp140 CFI was kindly provided by Dr. James Peacock (CAVD Duke University, Durham, NC, USA). The following reagents were obtained through the NIH HIV Reagent Program, Division of AIDS, NIAID, NIH: b12 IgG1 antibody, contributed by Dr. Dennis Burton and Dr. Carlos Barbas; anti-SIVmac gp120 monoclonal (KK8) antibody by Dr. Karen Kent, and anti-HIV p24 monoclonal (183-H12-5C) antibody contributed by Dr. Bruce Chesebro and Kathy Wehrly. All schemes were generated with BioRender.com.

## Author Contributions

**Conceptualization:** Christiane Stahl-Hennig, Klaus Überla.

**Funding acquisition:** Christiane Stahl-Hennig, Klaus Überla.

**Investigation:** Elie Richel, Arne Cordsmeier, Larissa Bauer, Kirsten Fraedrich, Ramona Vestweber, Berit Roshani, Nicole Stolte-Leeb, Armin Ensser.

**Methodology:** Elie Richel, Arne Cordsmeier, Larissa Bauer, Kirsten Fraedrich, Ramona Vestweber, Berit Roshani, Nicole Stolte-Leeb, Armin Ensser.

**Supervision:** Armin Ensser, Christiane Stahl-Hennig, Klaus Überla.

**Writing – original draft:** Elie Richel, Klaus Überla.

**Writing – review & editing:** Elie Richel, Arne Cordsmeier, Larissa Bauer, Kirsten Fraedrich, Ramona Vestweber, Berit Roshani, Nicole Stolte-Leeb, Armin Ensser, Christiane Stahl-Hennig, Klaus Überla.

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
