## [Decision Letter · Decision Letter 0]

28 Oct 2024

PPATHOGENS-D-24-02094Mechanisms of sterilizing immunity provided by an HIV-1 neutralizing antibody against mucosal infectionPLOS Pathogens Dear Dr. Überla, Thank you for submitting your manuscript to PLOS Pathogens. After careful consideration, we feel that it has merit but does not fully meet PLOS Pathogens's publication criteria as it currently stands. Therefore, we invite you to submit a revised version of the manuscript that addresses the points raised during the review process. Please submit your revised manuscript within 30 days Dec 27 2024 11:59PM. If you will need more time than this to complete your revisions, please reply to this message or contact the journal office at plospathogens@plos.org. Please include the following items when submitting your revised manuscript:*
A rebuttal letter that responds to each point raised by the editor and reviewer(s). You should upload this letter as a separate file labeled 'Response to Reviewers'. This file does not need to include responses to any formatting updates and technical items listed in the 'Journal Requirements' section below.*
A marked-up copy of your manuscript that highlights changes made to the original version. You should upload this as a separate file labeled 'Revised Manuscript with Track Changes'.*
An unmarked version of your revised paper without tracked changes. You should upload this as a separate file labeled 'Manuscript'. If you would like to make changes to your financial disclosure, competing interests statement, or data availability statement, please make these updates within the submission form at the time of resubmission. Guidelines for resubmitting your figure files are available below the reviewer comments at the end of this letter. We look forward to receiving your revised manuscript. Kind regards, David T. EvansAcademic EditorPLOS Pathogens Richard KoupSection EditorPLOS Pathogens Michael Malim

Editor-in-Chief

PLOS Pathogens

orcid.org/0000-0002-7699-2064 **Journal Requirements:** **Additional Editor Comments (if provided):****Reviewers' Comments:** Reviewer's Responses to Questions

**Part I - Summary**

Reviewer #1: Richel et al use an elegant SIV system to expand upon their work showing that Fc functions are not necessary for protective efficacy. The field has long discussed the levels of Env in relation to immune targeting and this work very nicely addresses that issue in respect to protection from different levels of challenge in macaques. The work is innovative and should be of interest to Plos Path readers. I have only minor comments.

1. The discussion seems unbalanced in that paragraphs describe how Fc functions may be useful (generally in rather contrived situations or using mice, generally with modest effects), yet this elegant study as well as multiple other well conducted primate studies with recent more potent Nab show that Fc functions are not needed. While it may be that non-nab exert some pressure on virus levels after infection has been established, this work along with other work of the authors and others, is convincing regarding the (lack of) role of non-nabs in prevention of infection.

2. Did the authors consider EM imaging of the SfdEnvLow, SfdEnvInter and

SfdEnvHigh viruses for Env spikes. I think that would add to the characterisation.

3. There are additional recent or relevant macaque papers that address this issue and are consistent with the authors data. PMID: 39198422 PMID: 34385004

Reviewer #2: This well written manuscript stems from the Stab et al Cell Rep Med article published by the Uberla group last year. In the current manuscript, Richel and collaborators tackle a very important but somehow controversial question in the HIV field: do Fc-effector functions contribute to bNAb mediated sterilizing protection from mucosal challenge? The authors slightly modified an ingenious and elegant system, first reported in the Stab et al CRM 2023 manuscript, where the challenge virus expresses a non-functional HIV Env mutant and a functional SIV Env. After reverse transcription, only the SIV Env is expressed and infected cells only express the SIV Env at their surface. Therefore, anti-HIV bNAbs (in the current work the author used PGT121 and its Fc-effector disabled variant) can only act on the oncoming virus, not on infected cells. In this well controlled and executed work, the authors conclude that Fc-effector functions of PGT121 do not contribute to bNAb mediated sterilizing immunity against mucosal viral infection.

Reviewer #3: This paper addresses the role of Fc-dependent antibody functions in preventing lentiviral infection. The authors previously demonstrated that neutralizing antibodies prevent viremia and seroconversion by blocking infection of the first cells. They also found that non-neutralizing mechanisms seemed to contribute to protection. However, they rightly point out a limitation of this latter finding, namely that the construct used contained an artificially high number of gp120 molecules. Work by Gach, et al. had shown that antibodies do not mediate antibody-dependent phagocytosis of authentic HIV-1 virions but do so when the amount of antigen on the virion surface is increased to unnatural levels. Richel, et al. now turn to confirming a role in vivo for Fc-dependent functions using viral constructs having a more natural number of Env spikes. Under such conditions, the neutralizing antibody PGT121 does not, in fact, utilize Fc-dependent antibody effector functions to prevent infection, consistent with the findings of Gach, et al.

**Part II – Major Issues: Key Experiments Required for Acceptance**

Reviewer #1: see above

Reviewer #2: The SIVdup system allows the authors to target the challenge virus at the beginning of the first replication cycle. Indeed, upon reverse transcription completion, the only Env being expressed is SIVEnv. Therefore, in their system, PGT121 can act only against viral particles during the first cycle, but not against infected cells. So, when the authors say that they explore “whether Fc-effector functions are relevant for providing sterilizing immunity against mucosal HIV infections” they must clearly specify which Fc-effector functions they refer to. They are certainly looking at virion opsonization, virolysis and perhaps ADCP, but certainly not ADCC which measures the lysis of the infected cells. In their system, the infected cells express SIVEnv and therefore it doesn’t allow them to look at ADCC mediated by the HIV-1 Env-specific PGT121 Ab. This needs to be clarified throughout the manuscript otherwise it would be misleading. It is possible that ADCC plays a role at the early stages upon infection, when there is a limited number of infected cells but their system doesn’t allow the authors to measure this.

There is no such a thing as non-neutralizing binding site for PGT121. In the context of the current work, PGT121 binds to an HIV Env which is inactive. I think that this is what the authors try to refer to when they say that it is non-neutralizing. However, whenever PGT121 interacts with its "binding site" it will neutralize if the HIV Env is fusion competent. This needs to be corrected otherwise it will confuse the reader.

Reviewer #3: (No Response)

**Part III – Minor Issues: Editorial and Data Presentation Modifications**

Reviewer #1: see above

Reviewer #2: In Table 1, the authors show the density of fd Env incorporated in the particles, but what is the density of SIV Env? Was it the same for the low, intermediate and high fd Env viruses? Could the SIV Env density interfere with the recruitment of effector cells after opsonization?

It would be informative if the authors document whether recognition of HIVEnv at the surface of the virus by the complement affects the viral entry mediated by SIVEnv?

Reviewer #3: 1. Many of the assays are done twice and “representative” results are shown. Why aren’t results of all the assays shown?

2. Secreted embryonic alkaline phosphatase should not be in caps.

3. Figure 2: Though stated elsewhere, it would be helpful to indicate in the legend what the abbreviations for the constructs refer to (e.g., SfdEnvLow = SIVdup pseudotyped with a low quantity of fusion-deficient HIV Env).

4. Figure 2A: Can the blot results be quantified?

5. Figure 2D: The figure is difficult to read because of the symbols and colors chosen.

6. Line 310: Should mention that the challenge is with SIVdup pseudotyped with the indicated Envs.

7. Line 335: Please clarify what is meant by “templates”.

8. Line 351: Did that animal have lower serum concentrations of mAb than others? Also, it would be best to show the relevant pharmacokinetic data for each animal in a supplementary figure or table.

9. Should mention the rationale for combining blood and lymph node results for the main comparison between groups.

10. The authors claim there is “sterilizing immunity”. But all the animals have some virus. Please clarify.

11. Line 413: the connection between the Hangarnter results and the “intrinsic characteristics of each single antibody” is rather vague. Please clarify.

PLOS authors have the option to publish the peer review history of their article (what does this mean?). If published, this will include your full peer review and any attached files.

Reviewer #1: No

Reviewer #2: No

Reviewer #3: **Yes: **Donald Forthal

---

## [Editor Report · Decision Letter 1]

26 Nov 2024

Dear Prof. Dr. Überla,

We are pleased to inform you that your manuscript 'Mechanisms of sterilizing immunity provided by an HIV-1 neutralizing antibody against mucosal infection' has been provisionally accepted for publication in PLOS Pathogens.

Best regards,

David T. Evans

Academic Editor

PLOS Pathogens

Richard Koup

Section Editor

PLOS Pathogens

Michael Malim

Editor-in-Chief

PLOS Pathogens

orcid.org/0000-0002-7699-2064
---

## [Editor Report · Acceptance letter]

6 Dec 2024

Dear Prof. Dr. Überla,

We are delighted to inform you that your manuscript, "Mechanisms of sterilizing immunity provided by an HIV-1 neutralizing antibody against mucosal infection," has been formally accepted for publication in PLOS Pathogens.

Best regards,

Sumita Bhaduri-McIntosh

Editor-in-Chief

PLOS Pathogens

orcid.org/0000-0003-2946-9497

Michael Malim

Editor-in-Chief

PLOS Pathogens

orcid.org/0000-0002-7699-2064